neuroscience, computational biology, cognition

cognitive neuroscience, pupillometry, information theory, computational neuroscience, psychopysiology, arousal

**Author for correspondence:**
Alexandre Zénon
e-mail: alexandre.zenon@u-bordeaux.fr

# Eye pupil signals information gain

Alexandre Zénon

INCIA, CNRS, Bordeaux, France

AZ, 0000-0001-7989-1261

In conditions of constant illumination, the eye pupil diameter indexes the modulation of arousal state and responds to a large breadth of cognitive processes, including mental effort, attention, surprise, decision processes, decision biases, value beliefs, uncertainty, volatility, exploitation/exploration trade-off, or learning rate. Here, I propose an information theoretic framework that has the potential to explain the ensemble of these findings as reflecting pupillary response to information processing. In short, updates of the brain's internal model, quantified formally as the Kullback–Leibler (KL) divergence between prior and posterior beliefs, would be the common denominator to all these instances of pupillary dilation to cognition. I show that stimulus presentation leads to pupillary response that is proportional to the amount of information the stimulus carries about itself and to the quantity of information it provides about other task variables. In the context of decision making, pupil dilation in relation to uncertainty is explained by the wandering of the evidence accumulation process, leading to large summed KL divergences. Finally, pupillary response to mental effort and variations in tonic pupil size are also formalized in terms of information theory. On the basis of this framework, I compare pupillary data from past studies to simple information-theoretic simulations of task designs and show good correspondance with data across studies. The present framework has the potential to unify the large set of results reported on pupillary dilation to cognition and to provide a theory to guide future research.

## 1. Cognitive pupillary response

Besides the well-known response of pupillary muscles to light, which narrows the range of light intensity reaching the retina and optimizing its information capacity [1], pupil size varies also as a function of a wealth of cognitive phenomena, including mental effort [2–5], surprise [6–15], emotion [16], decision processes [17–20], decision biases [19,21,22], value beliefs [23–25], volatility (unexpected uncertainty; [10,26–28]), exploitation/exploration trade-off [29,30], attention [31–36], uncertainty [12,19,21,23,25,37,38], confidence [39], response to reward [40], learning rate [10,41–43], neural gain [10,36,44,45], or urgency [46]. These variations in diameter follow coherent changes in neural activity throughout the cortex, regulated by neuromodulators, and is referred to as *arousal* [47–50]. The present work is based on the strong hypothesis that the ensemble of phenomena that trigger changes in pupil-linked arousal all depend on a basic underlying information theoretic process: the update of brain internal models. I will review a large breadth of findings from the literature and will reinterpret them under the light of that framework.

## 2. Surprise and self-information

One of the first cognitive variables that was shown to influence pupillary responses is surprise, defined in information theory as the negative logarithm of the probability of an event. This quantity is also called self-information, because it measures how much information is gained when observing an event. Pupil size has been shown to respond vigorously and robustly to

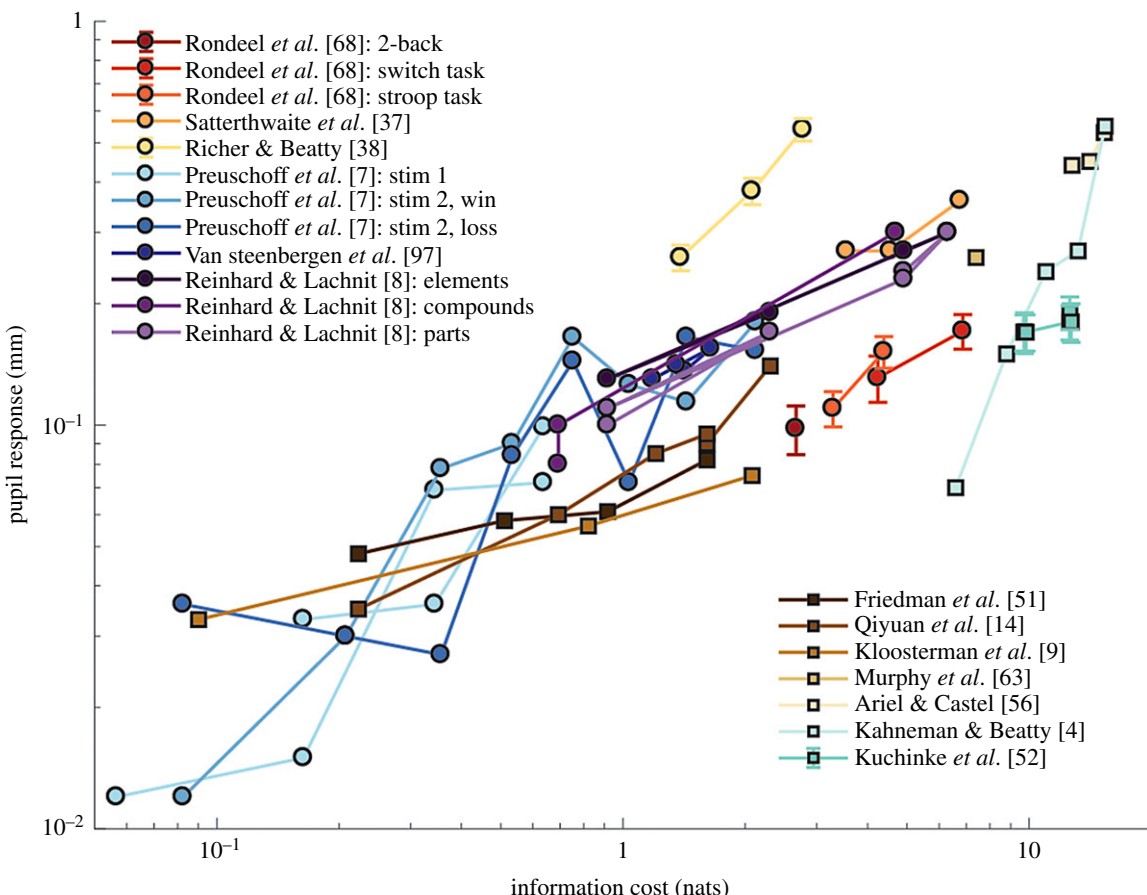

**Figure 1.** Relationship between information cost and pupil dilation in previous studies. Information cost was quantified as the KL divergence between prior and posterior beliefs. Squares in the graph illustrate pupillary responses to discrimination or detection tasks, in which KL divergence simplifies to stimulus self-information. Circles illustrate pupil dilations in response to task variables and decision making. See the electronic supplementary material for details. (Online version in colour.)

surprise, with dilation in response to events in inverse proportion to their frequency of occurrence in a trial [14,51,52]. The pupil also responds to stimulus disappearance, in inverse proportion to how likely the stimulus is to disappear at that given time [9]. Along the same line, pupillary dilation has been reported in relation to the probability of a reward outcome, independently of its sign (i.e. responses are equivalent for losses and rewards; [6,25,37]) or even to the occurrence of errors, as a function of their likelihood [53]. When events have probability distributions defined along continuous feature spaces (e.g. position, number line), the pupil also responds in inverse proportion to the probability density of occurrence of that feature [10,13]. When event occurrences depend on past trial history, pupil responses reflect surprise taking account of that history [10,12]. Despite this apparent consistency of findings, no attempts have been made so far to assess whether the relationship between pupil size and event probability follows a logarithmic trend, as predicted if pupil signals self-information. To step in this direction, the data from the aforementioned studies is plotted against quantified surprise values in figure 1 (see squares in the figure). This analysis is restricted to studies that reported probabilities quantitatively and measured pupil size in millimetres or per cent. These simulations show that pupil dilation is roughly linearly proportional to self-information, within and across studies. Precise comparison across studies is not possible given that detailed conditions are not available (i.e. time and performance pressure, lighting conditions, baseline arousal levels, etc.) and that measurement methods may differ. Therefore, the present results merely illustrate the plausibility of the hypothesis, but do not demonstrate it.

## 3. Information about task variables

The examples mentioned so far show that *pupil size dilates in proportion to the amount of information needed to encode sensory stimuli*. When a surprising stimulus is presented, self-information is large and pupils dilate. However, sensory stimuli such as cues, can also carry information about other, separate events. Pupillary response to such cases was investigated by Preuschoff et al. [7], in which stimuli informed participants on their winning probability. Subjects had to bet on which of two cards, whose values were revealed afterwards, was going to be larger. In this study, Preuschoff and colleagues looked at the pupil response to the display of the first card value. Here, all values (from 1 to 10) were equally likely, such that self-information was equal in all conditions. However, some cards provided more information than others about the chance of having a winning or losing bet. For example, when the first card was a 10, there was a guarantee of winning/losing if the participant had bet on the first card being larger/smaller (there were no ties in the game). Conversely, a 5 provided little information about the chance of winning, since probabilities were still close to 50–50. Such gradual gain of information about the probability distribution of a variable (chance of winning in the present case)

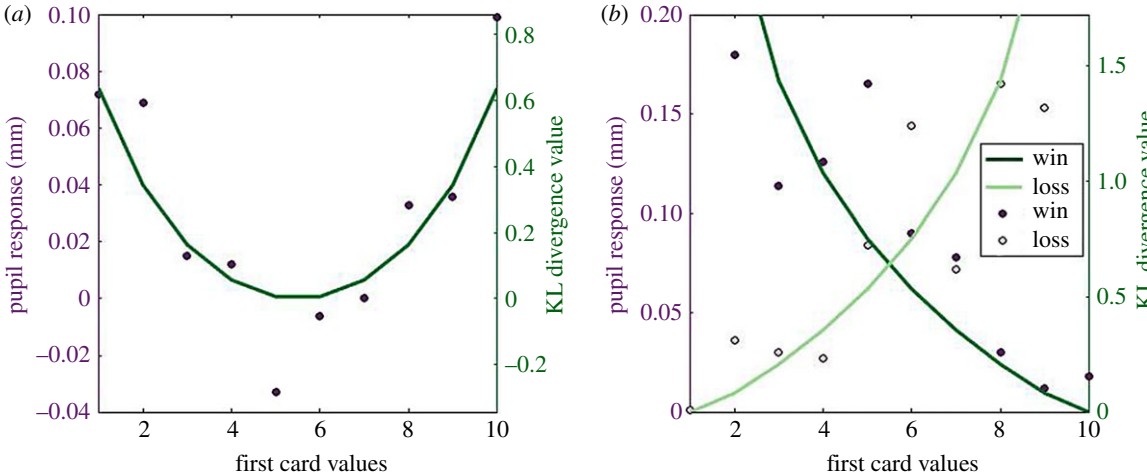

**Figure 2.** Data from Preuschoff *et al.* [7] (left *y*-axis), together with simulations based on the KL divergence between probability distribution of winning before and after viewing the stimuli (right *y*-axis). Responses to first card presentation are shown in (*a*), whereas (*b*) illustrates responses to second card presentation. See the electronic supplementary material for details. (Online version in colour.)

can be quantified by the Kullback–Leibler (KL) divergence between *prior* and *posterior* variable distributions. The KL divergence can be interpreted as the amount of information gained about the true probability distribution of a variable, after receiving new data. The KL divergence provides a generalized measure of information gain that is equivalent to self-information in the case of detection or discrimination tasks. Remarkably, the pupillary response to the first card value presentation in Preuschoff *et al.* [7] followed closely the KL divergence between subjects' belief on winning probability before and after observing the first card value (see light blue circles in figure 1 and figure 2*a*), even though these results were not discussed as such in the paper.

When the second card was presented, different situations could occur. The predictions could be confirmed, in which case little information would be gained (e.g. first card was 8, predicting first card being larger and second card was 5, confirming predictions), or they could be contradicted, in which case a lot of information would be gained (e.g. first card was 8 but second card was 9). Here again, the pupil responded in proportion to the amount of information being gained about winning probability, quantified as KL divergence (figures 1 and 2*b*). The findings of Preuschoff *et al.* [7] are compelling for several reasons. First, pupil size variations occurred following participants' choice and were thereby not affected by decision processes or motor responses, reflecting purely inferential processes. Second, they allow us to make clear quantitative predictions in terms of information processing and these predictions are strikingly confirmed.

One difference between surprise and KL divergence models of pupil response is that, *if the pupil responded only to surprise, it would always depend on the frequency of occurrence of presented stimulus, independently of task.* By contrast, KL divergence models predict that the pupil will respond to the amount of information provided by stimuli about task variables. This difference was exploited in two studies by Reinhard and co-workers [8,54] in which stimulus probabilities were manipulated in GO/NOGO tasks. In accordance with the information model, Reinhard *et al.* showed that pupillary response depended only on the probability of occurrence of the features of the GO/NOGO stimuli that were informative about the task (e.g. when GO was defined by the

occurrence of 1-letter as opposed to 2-letter stimuli, the identity of the letter being presented was irrelevant and failed to affect pupil response; see simulated results in figure 1). More generally, several studies have found that pupillary responses to stimuli depend on whether they are attended to or not [31,32,34,55,56] and that these responses scale with the subjective salience of the stimuli [35,56,57]. In attentional blink experiments, targets that closely follow previous target occurrences sometimes remain undetected. In these cases, pupillary response to target occurrence is greatly diminished [32]. Larger pupil dilation is associated with larger distractor interference [58], and increased processing of subliminal cues [59], in agreement with the view that pupil response scales with the quantity of visual information being processed.

## 4. Decision making

When decisions are made in the absence of uncertainty, such as in simple stimulus-response association tasks, the relationship between pupil response and information gain is straightforward. For example, in Richer and colleagues, both reaction time and pupil dilation were shown to vary as a function of the number of stimulus-response associations [38], in accordance with the classical Hick–Hyman law [60]. Here, the information cost of the decision can be quantified as the log of the number of possible stimulus-response associations in the task, which is equivalent to the KL divergence between prior and posterior beliefs [61] (see figure 1, yellow circles).

In conditions of uncertainty, the situation is slightly more complex. Satterthwaite and colleagues tested participants on a task similar to that of Preuschoff *et al.* [7], except that the decision followed, rather than preceded, the display of the first card value [37]. Participants had to pick either the face-up or face-down deck of cards. The second card value was then revealed and the trial was won if the card from the chosen deck was the largest [37]. Interestingly, in that case, the results were exactly opposite to those of Preuschoff *et al.*: when the first card was less informative (e.g. 5), making it more difficult for the subject to choose which deck to pick, the pupil response was larger than when the first number was either small or large, a case for which a decision was easier to make. The reaction time associated

*Proc. R. Soc. B* **286**: 20191593

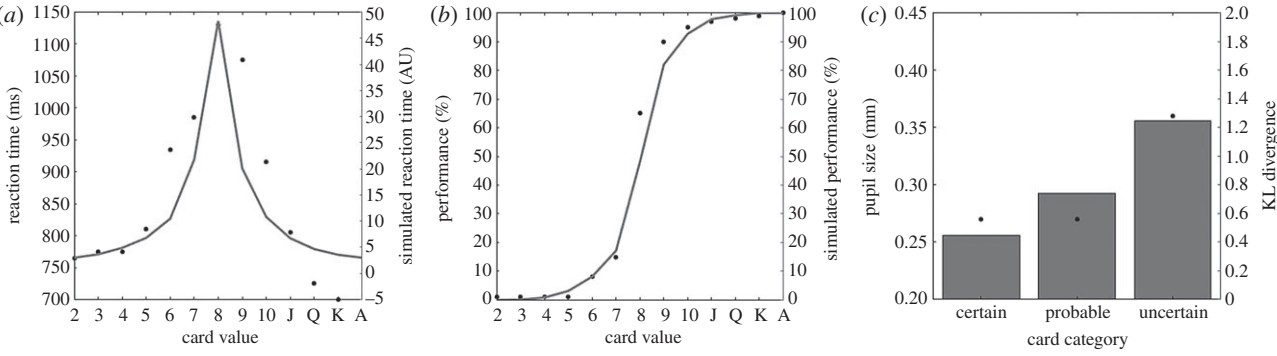

**Figure 3.** Simulation of reaction times (*a*) and per cent correct responses (*b*) from Satterthwaite *et al.* [37] by means of a drift diffusion process (DDM) process. (*c*) Illustrates the resulting KL divergences (grey bars), which follow the same trend (increasing with uncertainty) as the pupil size reported in the original study (black dots). It is noteworthy that the model used to simulate these data has the decision threshold as a single degree of freedom. See the electronic supplementary material for more details.

with the decision followed the same pattern, being larger for less informative values. This observed relationship between reaction time and pupillary dilation has been found in many studies [2,19,24,46,62–64] and pupillary responses are best modelled by means of regressors that extend during the whole reaction time period of the trial rather than by brief pulses limited to stimulus onset [18,22]. These findings suggest that the process from which pupillary dilation originates is maintained during the whole decision process.

The finding that uncertain or conflictual decisions are slower than decisions for which more information is available from stimulus is classical in the decision-making literature. It can be modelled as a drift diffusion process in which noisy evidence accumulates until a threshold is reached and in which the rate of accumulation depends on how close the option values are to each other [65,66]. Drift diffusion models can also be interpreted as time-resolved Bayesian decision-making processes in which each accumulation step corresponds to the update of prior to posterior belief [67]. The noisier the evidence, the more updates will tend to go in the wrong direction. Therefore, the summed quantity of information accumulated over the whole decision process is larger when evidence is noisy than when it is not. Thus, results from Satterthwaite *et al.* [37] can be accounted for by considering the sum of the KL divergences resulting from every update along the drift diffusion process (see figure 3 and light orange circles in figure 1). In Urai *et al.* [19] and Colizoli *et al.* [24], pupil size was measured during motion discrimination tasks and was shown to vary in parallel with decision uncertainty and reaction time: it decreased with stimulus strength for correct trials (low uncertainty), but increased with stimulus strength in error trials (high uncertainty). This pattern of results can also be explained by resorting to drift diffusion models of decision making and by assuming variable drift rates [66]. Along the same line, Cheadle *et al.* [45] showed that during a task in which evidence accumulated over eight successive stimulus presentations, pupillary responses were proportional to the amount of evidence provided by each stimulus. Moreover, this response was modulated by recency and confirmation biases, which both also affected decisions. So pupil responses tracked decision updates, as predicted by the present proposal. In de Gee *et al.* [18,22], pupil responses in detection and 2-alternative forced choice tasks were shown to be inversely proportional to the probability of the choice and hence to the KL divergence between prior and posterior: in conservative participants (biased towards NO), YES

choices led to larger responses, while the opposite tended to be found in more liberal participants (biased towards YES). Pupil responses were also shown to vary as a function of the influence of the prior on perceptual decisions in de Gee *et al.* [22] and Krishnamurthy *et al.* [21]: when prior beliefs have less weight (because of better control or attentional allocation or because of low prior reliability), more information is extracted from the sensory stimulus, KL divergence is larger, and the pupil dilates more. Along the same line, when the occurrence of surprising outcomes suggests the task structure may have changed, pupil dilation is even larger [10,21,26]. This is because such environmental volatility is associated with increased learning rate and thus increased influence of sensory evidence on internal models of the task. Indeed, the magnitude of the pupil response correlated with the extent to which volatility affected learning rate [10,21]. Together, these findings on pupillary response to volatility and surprise confirm that pupil diameter scales with how much novel sensory evidence is used to update current belief states.

# 5. Mental effort

Another common finding in the literature is that pupil size varies as a function of task demands and subject's engagement in the task, suggesting the view that pupillary dilation indexes mental effort [2,4,5,68–70]. We have recently proposed that mental effort too can be quantified as the average KL divergence between prior and posterior beliefs [61]. Effortful tasks often include a large number of associations between stimuli and responses, resulting in low prior beliefs for each association and requiring large updates in order to reach precise posterior beliefs (e.g. N-back task; see simulations of N-back task from Rondeel *et al.* [68] in figure 1, red circle). Other cases of difficult tasks are those in which prior beliefs do not match task statistics (e.g. Stroop task), or in which task statistics change constantly (e.g. switch tasks), also implying large updates and large information costs (see simulations of Stroop and switch tasks from Rondeel *et al.* [68] in figure 1, orange and yellow circles). So the present proposal that pupil size scales with information gain can also be applied to complex tasks and accounts for the classical relationship between mental effort and pupillary dilation.

## (a) Tonic pupil size

So far we have restricted our discussion to phasic pupil responses, i.e. the change in pupil size that follows event

onset. However, the tonic variations in pupillary diameter, usually measured during baseline epochs that precede trial onsets also have some interesting properties. These tonic pupillary changes have been related to the modes of discharge observed in noradrenergic neurons [29,30,47,48,50,63]. Large phasic responses occur when baseline firing rates of noradrenergic neurons are low and would correspond to small tonic pupil size, whereas large baseline noradrenergic activity would be associated with large tonic pupil size but small phasic responses [29,47,63,71,72]. Indeed, negative correlations between *spontaneous* changes in tonic and phasic pupil size have been reported repeatedly [18,20,29,44,63,72–74], even though *task-induced* or interindividual changes in tonic and phasic pupil size often go in the same direction [10,12,21,25,30,75].

The relationship between *spontaneous* changes in tonic pupil size and behaviour follows an inverted u-shape, with optimal performance being associated with intermediate pupil size, evoking Yerkes–Dodson Law [47,48,58,63,72]. Large tonic pupil sizes are concurrent with mind-wandering, distractibility, and exploratory behaviour [29,30,33,76,77], while very low tonic pupil sizes are associated with low vigilance and sleepiness [36,47,63,71,72,78–80]. However, in contrast with the aforementioned spontaneous changes, increases in tonic size that are *task-induced* occur, on the contrary, in conditions of high task demand: when taxing working memory [81], when counting stimuli silently [82], following changes of contingency [10,21,25,26,30,83], or in conditions of high uncertainty [12].

Assuming that tonic variations of pupil size, like phasic task-induced changes, reflect quantitatively the amount of information being processed by the brain may help reconcile these contradictory findings in a parsimonious way. When information is attached to an abrupt sensory signal, it leads to phasic dilation whose magnitude is proportional to the KL divergence between prior and posterior beliefs. In the absence of clear onset, tonic pupil size reflects information processing from memory, i.e. manipulation of working memory, planning, mind-wandering, mental imagery, or offline learning. This putative relationship between pupil dilation and information processing from memory may appear counterintuitive at first sight, because from the point of view of the individual, there is no 'novel' information when retrieving data from memory. However, processing data from memory involves inference and learning similar to those involved in sensory processing. In agreement with this view, mental imagery is known to recruit similar brain resources as sensory processing [84] and learning occurs even in the absence of sensory input [85]. Therefore, tonic pupil size would increase when cognitive activity occurs out of sync with task events [76], hence decreasing limited cognitive resources available for the main task [61], leading to distractibility and exploratory behaviour, but it would also increase during demanding covert computations on working memory [25,81,82]. However, confirming this hypothesis requires quantifying out-of-sync information processing in terms of KL divergence, like I did for phasic pupillary responses. Since I cannot provide such quantified predictions on the basis of current literature, this will have to rely on future experimental studies.

## 6. Relation to alternative theories

Pupillary responses, because of their relation to the noradrenergic system [71], have previously been linked to unexpected uncertainty [27,86], sometimes taken as a synonym to surprise [6,7,28] and sometimes as an equivalent of volatility, i.e. how likely the environment dynamics is to change [21,27,86–89]. These two definitions are strongly related since surprising observations suggest that the statistical structure of the environment may have changed [90]. While surprise is event-related and could be linked to phasic pupillary changes [28], volatility varies slowly and could be related to tonic pupil size [27]. Unexpected uncertainty relates also to the problem of exploitation/exploration trade-off, another concept linked to pupillary responses [29,30,83,91]: when confidence in the internal model of the environment drops following surprising observations, exploitation strategies lose value with respect to alternative exploration strategies [86]. However, recent data have shown that variations of tonic pupil size are not indicative of unexpected uncertainty, but are rather a signature of reducible uncertainty (ambiguity resulting from poor model of environment, caused by undersampling; [21]) or expected uncertainty (related to inherent noise; [12]). This is also in line with the finding that pupil size does not depend only on noradrenaline but also on other neuromodulators such as acetylcholine [50], whose function has been associated with encoding of expected uncertainty [27]. Phasic pupillary responses, on the contrary, were shown to correlate with unexpected uncertainty [21]. However, since volatility is a slow-changing property of the environment, this observed correlation with phasic pupillary changes must reflect the fact that, when prior knowledge on environment is unreliable (i.e. volatility is high), more weight is given to new sensory evidence, as opposed to prior biases [27,86,92], and model updates between prior and posterior beliefs are more expensive [92], leading to larger pupillary dilations. Overall, current evidence does not seem to favour the view that pupil dilation would be indicative of specific types of uncertainty but, as I argue in the present work, would rather signal information processing, which itself depends strongly on uncertainty conditions.

## 7. Limitations

Notably, two studies reported results that appear to be in contradiction with the present information model. In O'Reilly *et al.* [13], the onset of unexpected saccadic targets led to pupillary dilations, but when these violations of expectation indicated the need to update the internal model of saccade target distributions, pupillary responses were smaller than when these unexpected events were identified as outliers (identified by their colour). In Van Slooten *et al.* [23], pupillary response to the outcome of subjects' choices in a 2-arm bandit task was shown not to depend on modelled expectations: when subjects were thought to expect a large reward, their pupillary response was similar regardless of feedback. Further, the magnitude of the decision-related response scaled with the difference between the available options, and feedback pupillary response was inversely proportional to the model learning rate, both results being in apparent contradiction with the previous literature [10,19,37] and the present proposal. In both aforementioned cases, pupillary responses were compared to variables of computational models fitted to behaviour, as opposed to direct task variables. The conclusions drawn from these

models are valid only to the extent that their underlying assumptions are justified. For example, in O'Reilly et al. [13] the model assumed participants did not update their internal model when faced with outlier stimuli. However, it could be argued that participants always updated their internal models in the face of surprising targets but had to put extra work to cancel these updates when figuring out that the target was an outlier. So while the results of O'Reilly et al. [13] and Van Slooten et al. [23] appear to contradict our view and invite us to remain cautious in our conclusions, possible alternative interpretations of their data suggest that more investigations should be conducted to resolve this apparent inconsistency.

Another potential weakness of the present framework is that it assumes that pupil-linked arousal increases synchronously with information processing demands. However, some lines of evidence indicate that arousal may also increase in anticipation of demands [77,93]. It remains to be determined whether such anticipatory increases in arousal reflect model updating during preparatory processing, which would still be in agreement with the present model, or whether they point to a different time course in which arousal increases before cognitive processing takes place.

Finally, it should be mentioned that beside arousal, cognitive processing may also affect pupil size through its modulatory influence on the pupil light-reflex [70,94–96]. In some situations, this source of influence could intermingle with the arousal-related pupil changes addressed in the present work. Fully understanding the pupillary signal will thus require us to be able to account for these multiple factors.

## 8. Conclusion

In the present paper, the factors that trigger changes in pupil-linked arousal were discussed under the light of information theoretic framework. The hypothesis that pupil size scales with the amount of information being processed, allowed me to explain a wide range of data, sometimes with quantitative predictions. This view applies both to tonic and phasic pupillary responses, the difference being that phasic responses mark information processing triggered by precise event onset while tonic pupillary changes are not precisely aligned to external events. While the set of results discussed here is broadly in support with the proposed hypothesis, it is still far from being conclusive, and further experiments should be performed to attempt to refute the present proposal.

Beside the factors that trigger pupillary changes, an equally important issue concerns the computational effects of pupil-linked arousal, and more generally, its functional role in brain computations. This issue goes beyond the scope of the present paper and will be discussed in future work.

Data accessibility. This article has no additional data.
Competing interests. I declare I have no competing interests.
Funding. This research was funded by IdEx Bordeaux (Junior Chair) and Agence Nationale de la Recherche (ANR; JCJC grant).
Acknowledgements. I wish to thank Oleg Solopchuk, Stefano Ioannucci, and Sze-Ying Lam for their constructive comments on an earlier version of the manuscript.

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
