## [Reviewer comments · Proceedings of the Royal Society B: Biological Sciences]

Review History

RSPB-2019-1593.R0 (Original submission)

Review form: Reviewer 1

Recommendation

Major revision is needed (please make suggestions in comments)

Scientific importance: Is the manuscript an original and important contribution to its field?

Excellent

General interest: Is the paper of sufficient general interest?

Good

Quality of the paper: Is the overall quality of the paper suitable?

Acceptable

Is the length of the paper justified?

Yes

Should the paper be seen by a specialist statistical reviewer?

No

Do you have any concerns about statistical analyses in this paper? If so, please specify them explicitly in your report.

No

It is a condition of publication that authors make their supporting data, code and materials available - either as supplementary material or hosted in an external repository. Please rate, if applicable, the supporting data on the following criteria.

Is it accessible?

N/A

Is it clear?

N/A

Is it adequate?

N/A

Do you have any ethical concerns with this paper?

No

Comments to the Author

In this paper, the author suggests that 'updates of the brain [sic] internal model' quantified formally as the KL divergence between prior and posterior beliefs, are the common denominator explaining pupil dilation. He provides various sources of explicit justification for this - reanalysing some important datasets; and provides qualitative arguments in a number of other cases.

I think that this is an interesting and bold claim - and there is certainly an element of truth in that *one* of the drivers of pupil dilation (and arousal) might indeed well be model updating. However, the quantification is complicated (even in fig 2, for one single expt, the mapping between pupil dilation and KL divergence is different for left and right plots, representing two informational points in the task), and the extension to dilation caused by purely internal processing (for instance when subjects think about something with strong emotional content) is unconvincing. The trouble with the latter is that there is no actual new information - so semantic contortions are necessary to shoe-horn the results into the hypothesis ('information processing from memory' - when there is no actual new information at all). One could say something similar for arousal associated with the (potentially future) need for motor effort; this is not so plausibly anything to do with changes to internal models. Even something like the Stroop task requires special 'pleading', in that the *information* in a Bayesian sense, is all available and the prior refuted at the time that the subject knows they have to ignore the written word. The difficulty is the *computation* required - and despite the loose term 'information processing', one cannot use Shannon-like information theory as a way of couching all computations.

In sum, pupil dilation, like arousal itself, it really responding to a variety of demands - and the priority of this particular interpretation is not so convincing.

Nevertheless, I think that the author's re-analysis of the various datasets is interesting - and would encourage him to reformulate the paper without the most all-encompassing claim, and concentrate instead more narrowly on these.

more minor points

- it's neuromodulators rather than neuromediators

- it's a bit unfair to use a log-log plot for a measure such as information cost, which is *already* in logarithmic form. Log-log plots can hide a multitude of sins - this should be shown in linear terms.

Review form: Reviewer 2

Recommendation

Accept with minor revision (please list in comments)

Scientific importance: Is the manuscript an original and important contribution to its field?

Good

General interest: Is the paper of sufficient general interest?

Excellent

Quality of the paper: Is the overall quality of the paper suitable?

Acceptable

Is the length of the paper justified?

Yes

Should the paper be seen by a specialist statistical reviewer?

No

Do you have any concerns about statistical analyses in this paper? If so, please specify them explicitly in your report.

No

It is a condition of publication that authors make their supporting data, code and materials available - either as supplementary material or hosted in an external repository. Please rate, if applicable, the supporting data on the following criteria.

Is it accessible?

N/A

Is it clear?

N/A

Is it adequate?

Yes

Do you have any ethical concerns with this paper?

No

Comments to the Author

"Eye pupil signals information gain"

The manuscript deals with pupil size changes in response to mental effort (or perturbation in

neural activity in the cortex) that results from cognitive processes. The corresponding pupil changes are usually dilations and occur slowly over several seconds, even under conditions of relatively steady lighting. The author proposes that the processing of information in the cortex can slowly disturb the cortical pathways that can regulate the control of pupil size with subsequent dilations and constrictions of the pupil. Decision making involves mental effort and poor information or uncertainty modulates mental loads which in turn can affect pupil size. All this is known and reasonable.

The author produces a convincing analysis of existing data and proposes that divergence between prior and posterior beliefs modulates mental effort in decision making and that this cognitive effort is reflected in the pupil. He goes on to say that the pupil responses involved are 'proportional to the amount of information the stimulus carries about itself and to the quantity of information it provides about other task variables'. This is also reusable and the latter part may also be original.

Although this statement captures almost any aspect of cognitive effort triggered by internal or external stimuli, often visual, the author makes no reference to many other studies when perturbations in neural activity that relate specifically to the amount of information carried by the stimulus also trigger changes in pupil size. The latter are largely transient constrictions with latencies that reflect the cortical processing of different stimulus attributes (Proc Biol Sci. 1998; 265(1412): 2321-5).

I wonder if the all-embracing framework proposed by the author has the potential to also account for the stimulus specific information that causes constrictions of the pupil? It seems paradoxical 'that pupil size dilates in proportion to the amount of information needed to encode sensory stimuli', as stated by the author, yet it initially constricts in proportion to the amount of information carried by other sensory stimuli.

The manuscript covers many important aspects of cognitive processing and lists 90 references! The latter illustrates the extent and interest in this topic and the number of studies that involved the measurement of pupil size changes triggered by changes in mental effort. In this respect, the manuscript makes a significant contribution in attempting to bring together a number of related mental demands that cause dilation of the pupil. The information theory arguments put forward by the author are plausible, but further (experimental) studies are needed to isolate specific pupil responses so to establish whether pupil size truly scales with the amount of information involved and why stimulus specific information has the opposite effect by causing constrictions of the pupil that are proportional to the information content of the stimulus.

Decision letter (RSPB-2019-1593.R0)

05-Aug-2019

Dear Dr Zénon:

Your manuscript has now been peer reviewed and the reviewers' comments (not including confidential comments to the Editor) are included at the end of this email for your reference. As you will see, while positive, the reviewers have raised some concerns with your manuscript and we would like to invite you to revise your manuscript to address them.

We do not allow multiple rounds of revision so we urge you to make every effort to fully address all of the comments at this stage. If deemed necessary by the Associate Editor, your manuscript will be sent back to one or more of the original reviewers for assessment. If the original reviewers

are not available we may invite new reviewers. Please note that we cannot guarantee eventual acceptance of your manuscript at this stage.

Research ethics:

Use of animals and field studies:

Please submit a copy of your revised paper within three weeks. If we do not hear from you within this time your manuscript will be rejected. If you are unable to meet this deadline please let us know as soon as possible, as we may be able to grant a short extension.

Best wishes,
Innes Cuthill,
Reviews Editor
mailto:proceedingsb@royalsociety.org

Reviewer(s)' Comments to Author:

Referee: 1

Comments to the Author(s)

In this paper, the author suggests that 'updates of the brain [sic] internal model' quantified formally as the KL divergence between prior and posterior beliefs, are the common denominator explaining pupil dilation. He provides various sources of explicit justification for this - reanalysing some important datasets; and provides qualitative arguments in a number of other cases.

I think that this is an interesting and bold claim - and there is certainly an element of truth in that *one* of the drivers of pupil dilation (and arousal) might indeed well be model updating. However, the quantification is complicated (even in fig 2, for one single expt, the mapping between pupil dilation and KL divergence is different for left and right plots, representing two informational points in the task), and the extension to dilation caused by purely internal processing (for instance when subjects think about something with strong emotional content) is unconvincing. The trouble with the latter is that there is no actual new information - so semantic contortions are necessary to shoe-horn the results into the hypothesis ('information processing from memory' - when there is no actual new information at all). One could say something similar for arousal associated with the (potentially future) need for motor effort; this is not so plausibly anything to do with changes to internal models. Even something like the Stroop task requires

special 'pleading', in that the *information* in a Bayesian sense, is all available and the prior refuted at the time that the subject knows they have to ignore the written word. The difficulty is the *computation* required - and despite the loose term 'information processing', one cannot use Shannon-like information theory as a way of couching all computations.

In sum, pupil dilation, like arousal itself, it really responding to a variety of demands - and the priority of this particular interpretation is not so convincing.

Nevertheless, I think that the author's re-analysis of the various datasets is interesting - and would encourage him to reformulate the paper without the most all-encompassing claim, and concentrate instead more narrowly on these.

more minor points

- it's neuromodulators rather than neuromediators

- it's a bit unfair to use a log-log plot for a measure such as information cost, which is *already* in logarithmic form. Log-log plots can hide a multitude of sins - this should be shown in linear terms.

Referee: 2

Comments to the Author(s)

"Eye pupil signals information gain"

The manuscript deals with pupil size changes in response to mental effort (or perturbation in neural activity in the cortex) that results from cognitive processes. The corresponding pupil changes are usually dilations and occur slowly over several seconds, even under conditions of relatively steady lighting. The author proposes that the processing of information in the cortex can slowly disturb the cortical pathways that can regulate the control of pupil size with subsequent dilations and constrictions of the pupil. Decision making involves mental effort and poor information or uncertainty modulates mental loads which in turn can affect pupil size. All this is known and reasonable.

The author produces a convincing analysis of existing data and proposes that divergence between prior and posterior beliefs modulates mental effort in decision making and that this cognitive effort is reflected in the pupil. He goes on to say that the pupil responses involved are 'proportional to the amount of information the stimulus carries about itself and to the quantity of information it provides about other task variables'. This is also reusable and the latter part may also be original.

Although this statement captures almost any aspect of cognitive effort triggered by internal or external stimuli, often visual, the author makes no reference to many other studies when perturbations in neural activity that relate specifically to the amount of information carried by the stimulus also trigger changes in pupil size. The latter are largely transient constrictions with latencies that reflect the cortical processing of different stimulus attributes (Proc Biol Sci. 1998; 265(1412): 2321-5).

I wonder if the all-embracing framework proposed by the author has the potential to also account for the stimulus specific information that causes constrictions of the pupil? It seems paradoxical 'that pupil size dilates in proportion to the amount of information needed to encode sensory stimuli', as stated by the author, yet it initially constricts in proportion to the amount of information carried by other sensory stimuli.

The manuscript covers many important aspects of cognitive processing and lists 90 references!

The latter illustrates the extent and interest in this topic and the number of studies that involved the measurement of pupil size changes triggered by changes in mental effort. In this respect, the manuscript makes a significant contribution in attempting to bring together a number of related mental demands that cause dilation of the pupil. The information theory arguments put forward by the author are plausible, but further (experimental) studies are needed to isolate specific pupil responses so to establish whether pupil size truly scales with the amount of information involved and why stimulus specific information has the opposite effect by causing constrictions of the pupil that are proportional to the information content of the stimulus.

Author's Response to Decision Letter for (RSPB-2019-1593.R0)

See Appendix A.

RSPB-2019-1593.R1 (Revision)

Review form: Reviewer 1

Recommendation

Accept with minor revision (please list in comments)

Scientific importance: Is the manuscript an original and important contribution to its field?

Good

General interest: Is the paper of sufficient general interest?

Good

Quality of the paper: Is the overall quality of the paper suitable?

Good

Is the length of the paper justified?

Yes

Should the paper be seen by a specialist statistical reviewer?

No

Do you have any concerns about statistical analyses in this paper? If so, please specify them explicitly in your report.

No

It is a condition of publication that authors make their supporting data, code and materials available - either as supplementary material or hosted in an external repository. Please rate, if applicable, the supporting data on the following criteria.

Is it accessible?

N/A

Is it clear?

N/A

Is it adequate?

N/A

Do you have any ethical concerns with this paper?

No

Comments to the Author

I remain a bit sceptical about the underlying purported mandatory relationship between information and computation (in the author's attempt to shoehorn effort/load into Shannon terms) - and the reply seems to me to flirt with the tautology that any computation is precisely identifiable with the change in information it produces (which makes the hypothesis lack interesting specificity). Indeed, Friston has himself in my hearing declared various of his theories to be tautologies - which takes them out of the realm of scientific inquiry.

However, the current paper is clear about the nature and status of its key hypothesis, and I wouldn't stand in its way. The title is now misleading - since the claim is about updating internal models, not gaining information.

Decision letter (RSPB-2019-1593.R1)

20-Aug-2019

Dear Dr Zénon

I am pleased to inform you that your manuscript RSPB-2019-1593.R1 entitled "Eye pupil signals information gain" has been accepted for publication in Proceedings B.

The referee is still somewhat sceptical about the hypothesis, but is happy that the idea gets "out there" and so has recommended publication. However, the referee suggests that the title needs changing. Therefore, I invite you to respond to the referee's comments and upload the final version of your manuscript. Because the schedule for publication is very tight, it is a condition of publication that you submit the revised version of your manuscript within 7 days. If you do not think you will be able to meet this date please let us know.

- DNA sequences: Genbank accessions F234391-F234402
- Phylogenetic data: TreeBASE accession number S9123
- Final DNA sequence assembly uploaded as online supplementary material
- Climate data and MaxEnt input files: Dryad doi:10.5521/dryad.12311

[http://datadryad.org/submit?journalID=RSPB&manu=\(Document not available\)](http://datadryad.org/submit?journalID=RSPB&manu=(Document%20not%20available)) which will take you to your unique entry in the Dryad repository. If you have already submitted your data to dryad you can make any necessary revisions to your dataset by following the above link. Please see <https://royalsociety.org/journals/ethics-policies/data-sharing-mining/> for more details.

Sincerely,

Editor, Proceedings B
mailto: proceedingsb@royalsociety.org

Associate Editor

Reviewer(s)' Comments to Author:

Referee: 1

Comments to the Author(s)

I remain a bit sceptical about the underlying purported mandatory relationship between information and computation (in the author's attempt to shoehorn effort/load into Shannon terms) - and the reply seems to me to flirt with the tautology that any computation is precisely identifiable with the change in information it produces (which makes the hypothesis lack interesting specificity). Indeed, Friston has himself in my hearing declared various of his theories to be tautologies - which takes them out of the realm of scientific inquiry.

However, the current paper is clear about the nature and status of its key hypothesis, and I wouldn't stand in its way. The title is now misleading - since the claim is about updating internal models, not gaining information.

Author's Response to Decision Letter for (RSPB-2019-1593.R1)

See Appendix B.

Decision letter (RSPB-2019-1593.R2)

28-Aug-2019

Dear Dr Zénon

I am pleased to inform you that your manuscript entitled "Eye pupil signals information gain" has been accepted for publication in Proceedings B.

If you are likely to be away from e-mail contact during this period, let us know. Due to rapid publication and an extremely tight schedule, if comments are not received, we may publish the paper as it stands.

Open access

You are invited to opt for open access via our author pays publishing model. Payment of open access fees will enable your article to be made freely available via the Royal Society website as soon as it is ready for publication. For more information about open access publishing please visit our website at http://royalsocietypublishing.org/site/authors/open_access.xhtml.

The open access fee is £1,700 per article (plus VAT for authors within the EU). If you wish to opt for open access then please let us know as soon as possible.

Paper charges

Sincerely,

Proceedings B

Appendix A

Revision of Manuscript ID RSPB-2019-1593, “Eye pupil signal information gain”

I want to thank the Referees for their constructive and overall positive assessment on the paper. Please find below my detailed responses to their comments followed by the corrected manuscript with changes highlighted in red.

Referee 1:

In this paper, the author suggests that 'updates of the brain [sic] internal model' quantified formally as the KL divergence between prior and posterior beliefs, are the common denominator explaining pupil dilation. He provides various sources of explicit justification for this - reanalysing some important datasets; and provides qualitative arguments in a number of other cases.

*I think that this is an interesting and bold claim - and there is certainly an element of truth in that *one* of the drivers of pupil dilation (and arousal) might indeed well be model updating. However, the quantification is complicated (even in fig 2, for one single expt, the mapping between pupil dilation and KL divergence is different for left and right plots, representing two informational points in the task).*

Indeed quantification across studies is difficult and the goal of the simulations I provide is merely to substantiate the plausibility of the hypothesis, not to demonstrate it, as now stated more clearly in the manuscript (lines 32-36).

I want to thank the Referee for pointing out the differences in the relation of proportionality between panels A and B of Figure 2. This has been addressed by applying a more systematic approach to fitting the data from Preuschoff et al. I now do so by fitting the relation of proportionality (which is the only parameter of the model) to the data by minimizing mean squared error (MSE). I then compared the corresponding AIC with the one obtained from the original model proposed in Preuschoff's paper. Results show that despite comparable quality of fit, the KL model resulted in better AIC value because of its lower complexity. This is now detailed in supplementary material.

The extension to dilation caused by purely internal processing (for instance when subjects think about something with strong emotional content) is unconvincing. The trouble with the latter is that there is no actual new information - so semantic contortions are necessary to shoe-horn the results into the hypothesis ('information processing from memory' - when there is no actual new information at all).

This is a fair point. However, information in the present work does not refer only to new data from the individual's perspective but is measured in terms of update of internal representations. Processing information from memory entails somehow replacing sensory entries with stored content (at any level in the processing hierarchy). This does not prevent later stages of processing to perform inference or even learning (e.g. consolidation), thereby leading to representation updates and, according to the hypothesis, arousal-linked pupil dilation equivalent to those observed during sensory processing. I have tried to make this point clearer in the manuscript (lines 176-181).

One could say something similar for arousal associated with the (potentially future) need for motor effort; this is not so plausibly anything to do with changes to internal models.

On the one hand, I agree with the Referee that it is possible that arousal could increase not only when faced with actual cognitive demands, but also in anticipation of such demand, especially when latency is critical. Indeed, this possibility is not accounted for by the present model and this limitation is now discussed in the paper (lines 233-238).

On the other hand, I disagree with the claim that such arousal response cannot be plausibly associated to updates of internal models. On the contrary, in the example provided by the Referee, a lot of cognitive processing has to be performed in anticipation of, and during physical effort. These computations can be easily framed in terms of internal model updates (e.g. Brown, H., Friston, K., Bestmann, S., 2011. Active inference, attention, and motor preparation. *Front. Psychol.* doi:10.3389/fpsyg.2011.00218).

*Even something like the Stroop task requires special 'pleading', in that the *information* in a Bayesian sense, is all available and the prior refuted at the time that the subject knows they have to ignore the written word. The difficulty is the *computation* required - and despite the loose term 'information processing', one cannot use Shannon-like information theory as a way of couching all computations.*

I do not agree with the premise that the prior should necessarily be refuted by task instructions. Priors should account for general statistical properties of the environment. If word naming is a very frequent stimulus-response association in natural environment, then it makes sense for its prior probability to be large and relatively inflexible: it would take a long time for a task not involving word-naming to undo this preference. The more training a given association has received, the less flexible and sensitive to learning the prior should be (i.e. habitual behaviour). At the extreme of this spectrum, priors on interoceptive variables could be hardly modifiable at all (see Zénon et al, 2018, *Neuropsychologia* for more discussion on this topic).

If one accepts the assumptions of the “Bayesian brain”, then all brain computations must implement updates on probabilistic representations of environment and task variables, which can be quantified by means of KL divergence, and this approach can be applied to Stroop task as well, without any special pleading. While I agree that some computations do not involve change in information content and can thus not be described through information theory (e.g. coordinate change), this is not the case in the Stroop task in which the extra cost of performing colour-naming is caused by the interference with the more salient word-naming process. Such case is especially well described in terms of prior-induced bias in Bayesian frameworks.

In sum, pupil dilation, like arousal itself, is really responding to a variety of demands - and the priority of this particular interpretation is not so convincing.

Nevertheless, I think that the author's re-analysis of the various datasets is interesting - and would encourage him to reformulate the paper without the most all-encompassing claim, and concentrate instead more narrowly on these.

In agreement with the Referee, I cannot claim that pupil-linked arousal is driven only by information gain, because at this stage, evidence is still too indirect and weak to be fully convincing. However, what I am trying to stress in the paper is that the proposed hypothesis is coarsely in agreement with existing data. Since it is also plausible and parsimonious, I consider it to be a valid hypothesis which we should now try to refute with experiments. I now mention this point in the conclusion (lines 249-251).

more minor points

- it's neuromodulators rather than neuromediators

This typo has been corrected.

*- it's a bit unfair to use a log-log plot for a measure such as information cost, which is *already* in logarithmic form. Log-log plots can hide a multitude of sins - this should be shown in linear terms.*

This figure has been presented in log-log scale because of the wide disparity of magnitudes across experiments. A linear scale tends to concentrate all points of the extreme lower left of the plot and makes it hard to read. Therefore, I have chosen to keep the original version of the plot. Naturally, I would be willing to use the linear version of the figure if it was deemed crucial by the Referee and/or the Editor.

Referee: 2

The manuscript deals with pupil size changes in response to mental effort (or perturbation in neural activity in the cortex) that results from cognitive processes. The corresponding pupil changes are usually dilations and occur slowly over several seconds, even under conditions of relatively steady lighting. The author proposes that the processing of information in the cortex can slowly disturb the cortical pathways that can regulate the control of pupil size with subsequent dilations and constrictions of the pupil. Decision making involves mental effort and poor information or uncertainty modulates mental loads which in turn can affect pupil size. All this is known and reasonable. The author produces a convincing analysis of existing data and proposes that divergence between prior and posterior beliefs modulates mental effort in decision making and that this cognitive effort is reflected in the pupil. He goes on to say that the pupil responses involved are 'proportional to the amount of information the stimulus carries about itself and to the quantity of information it provides about other task variables'. This is also reusable and the latter part may also be original. Although this statement captures almost any aspect of cognitive effort triggered by internal or external stimuli, often visual, the author makes no reference to many other studies when perturbations in neural activity that relate specifically to the amount of information carried by the stimulus also trigger changes in pupil size. The latter are largely transient constrictions with latencies that reflect the cortical processing of different stimulus attributes (Proc Biol Sci. 1998; 265(1412): 2321-5).

I wonder if the all-embracing framework proposed by the author has the potential to also account for the stimulus specific information that causes constrictions of the pupil? It seems paradoxical 'that pupil size dilates in proportion to the amount of information needed to encode sensory stimuli', as stated by the author, yet it initially constricts in proportion to the amount of information carried by other sensory stimuli.

In the study referenced by the Referee, pupillary constriction is caused by the onset of the visual stimulus, triggering a pupillary light reflex. I found no evidence that this constriction is proportional to the amount of information provided by the stimulus but it is definitely modulated by stimulus properties and stage of cortical processing. This is in agreement with other experimental evidence showing that the light reflex depends indeed in part on inputs from cortex. These findings are not contradictory, but complementary to our proposal, which tackles arousal-linked pupillary dilation triggered by cognitive processing. I now bring up these elements in the limitations section of the paper (lines 239-242).

The manuscript covers many important aspects of cognitive processing and lists 90 references! The latter illustrates the extent and interest in this topic and the number of studies that involved the measurement of pupil size changes triggered by changes in mental effort. In this respect, the manuscript makes a significant contribution in attempting to bring together a number of related mental demands that cause dilation of the pupil. The information theory arguments put forward by the author are plausible, but further (experimental) studies are needed to isolate specific pupil responses so to establish whether pupil size truly scales with the amount of information involved and why stimulus specific information has the opposite effect by causing constrictions of the pupil that are proportional to the information content of the stimulus.

I fully agree with the Referee regarding the need for more experimental work aimed at supporting or refuting the hypothesis put forth in the present paper. This is now mentioned in the paper (lines 249-251).

Appendix B

Revision of Manuscript ID RSPB-2019-1593, “Eye pupil signal information gain”

Please find below my detailed responses to the last comments of Referee 1.

Referee 1:

I remain a bit sceptical about the underlying purported mandatory relationship between information and computation (in the author's attempt to shoehorn effort/load into Shannon terms) - and the reply seems to me to flirt with the tautology that any computation is precisely identifiable with the change in information it produces (which makes the hypothesis lack interesting specificity). Indeed, Friston has himself in my hearing declared various of his theories to be tautologies - which takes them out of the realm of scientific inquiry.

This comment concerns more specifically the work that was published in an earlier paper (Zénon et al. 2018, *Neuropsychologia*) and on which the present work is partly based. In fact relating computation to information theory is a complex matter that is the topic of the whole field of algorithmic information theory. I am not claiming that “any computation is precisely identifiable with the change of information it produces”. More specifically, I propose to extend the “efficient coding” hypothesis by stating that the effort/load associated with a cognitive task is a function of its information cost, measured as the KL divergence between prior and posterior beliefs represented in the brain. This hypothesis, in contrast to Referee’s claim, results in clear, quantitative and testable predictions, which I detail in both the present and past papers.

However, the current paper is clear about the nature and status of its key hypothesis, and I wouldn't stand in its way. The title is now misleading - since the claim is about updating internal models, not gaining information.

Given that updating beliefs represented in internal models is the only way in which information can be gained by the system, I would like to maintain the current title unchanged, since it gives a better idea of the information theoretic perspective taken by the paper. Again, I am willing to compromise if it is deemed essential by the Referee or the Editor.